# Substantial Impairment of Quality of Life during COVID-19 Pandemic in Patients with Advanced Rectal Cancer

**DOI:** 10.3390/healthcare10081513

**Published:** 2022-08-11

**Authors:** Irene Dennison, Claudia Schweizer, Tim Fitz, Daniel Blasko, Christopher Sörgel, Annett Kallies, Linda Schmidt, Rainer Fietkau, Luitpold Valentin Distel

**Affiliations:** 1Department of Radiation Oncology, Universitätsklinikum Erlangen, Friedrich-Alexander-Universität Erlangen-Nürnberg (FAU), Universitätsstraße 27, 91054 Erlangen, Germany; 2Comprehensive Cancer Center Erlangen-EMN (CCC ER-EMN), Universitätsklinikum Erlangen, Friedrich-Alexander-Universität Erlangen-Nürnberg (FAU), 91054 Erlangen, Germany

**Keywords:** quality of life, COVID-19, pandemic, rectal cancer, radiochemotherapy, sex

## Abstract

The aim of this study was to analyze the quality of life of patients with advanced rectal cancer before and during the COVID-19 pandemic and to determine whether the pandemic affected patients’ quality of life. The study included 389 patients and was performed from May 2010 to June 2021. The fifteen months from March 2020 to June 2021 were categorized as the COVID-19 period. Patients were surveyed using the QLQ-C30 and QLQ-CR38 questionnaires. The questionnaires were used at different phases of radiochemotherapy: prior to RCT (day 1), during RCT (day 14), at the end of RCT (day 35), and prior to mesorectal surgery (day 70). Scores were formed from the questions. In addition, scores were analyzed for different age groups (<64 and >64) and sexes (female and male). Overall, patients reported lower functional scores and higher symptom scores during the pandemic than before the pandemic. Although it had been expected that older and younger patients would differ clearly, there were only minor differences. The comparison between the two sexes showed very different scores, with female patients having lower functional scores and higher symptom scores than male patients before and especially during COVID-19. In conclusion, age does not play a major role in quality of life, but sex does play an important role in perception of functioning and symptoms. COVID-19 also had a major impact on patients’ lives, as it was a very isolating and stressful time for everyone, especially cancer patients, which was reflected in worsening scores.

## 1. Introduction

In December 2019, the whole world was surprised by the first COVID-19 cases. In March 2020, the World Health Organization declared the presence of a pandemic [1]. Hospitals were flooded with patients. During this time, the German Ministry of Health announced that all surgeries in all hospitals that were not emergencies would be postponed. Experts decided that interdisciplinary management was needed for all cancer patients because resources were very limited [2]. In addition, strict access controls were implemented for visitors to the hospital. At times, no visitors were allowed. Most of the time, only one specific person was allowed to visit a patient at a time. In addition, visiting hours were limited.

Colorectal cancer is the third-most common cancer in Germany, with an incidence of 58,000 for both sexes and all age groups [3]. While colon cancer is treated only internally or surgically, advanced rectal cancer is usually treated with neoadjuvant radiochemotherapy followed by surgery. The goal of radiochemotherapy is to inactivate cancer cells and shrink the tumor before patients undergo surgery. The goal of this treatment is to protect the sphincter and preserve normal bowel function [4].

Neoadjuvant treatment has made several advances as it focuses on metastatic and locally advanced colorectal cancer. The SARS-CoV-2 infections were highly infectious and risky, especially for people and patients with other diseases and weak immune systems, including colorectal cancer patients, for whom the risk was much higher [1]. All of these limitations and risk factors could affect the quality of life in various aspects of patients’ lives. In this study, we compared and contrasted the quality of life of patients during the pandemic and prior to the pandemic. We were particularly interested in whether and how the pandemic had an impact on patients’ quality of life. In addition, the goal was to identify whether sex or age plays a special role in the pandemic situation and what would need to be considered in similar situations in the future.

## 2. Materials and Methods

Patient Cohort:

This study was conducted from May 2010 to June 2021 with a total of 389 patients who agreed to participate, focusing on March 2020 to June 2021, the months during COVID-19. There was a total of 350 patients prior to COVID-19 and 39 patients during the COVID-19 pandemic, including 247 male patients and 103 female patients prior to the pandemic and 23 male patients and 16 female patients during the pandemic. Patient characteristics are shown in Table 1, indicating the number of patients, sex, age group, tumor grade, number of patients without surgery, and different tumor stages. All patients who participated in the study were from Erlangen University Hospital, and the criteria for patient selection were advanced rectal cancer and neoadjuvant radiochemotherapy. Because only neoadjuvant treatments are performed at the institution, only these patients were included in the study.

Patient-Reported Outcome:

The European Organization for Research Treatment of Cancer (EORTC) QLQ-C30 and QLQ-CR38 questionnaires were used for the surveys. The QLQ-C30 questionnaire consists of 30 questions on a Likert scale with four to seven response options for each item. The QLQ-CR38 questionnaire consists of 38 questions, each with four to five response options. From these 68 questions (both questionnaires together), 27 scores were formed. The 27 scores were divided into function scores and symptom scores and expressed in percentage scales (0–100%). A higher value for the functional scores means that the patient feels good and/or strong in that area, whereas a lower value means that the patient feels weak and/or poor in that area. A higher value for the symptom score means that the patient feels bad and/or weak. A lower value, on the other hand, means that the patient has fewer problems and feels good. Questionnaires were given to patients at baseline (day-1), during RCT (day 12; week 2), at the end of RCT (day 35; week 5), and just before their surgery (day 70; week 10). Patients could complete the questionnaires during their follow-up sessions, or they were mailed directly to them. The results were digitized and stored using Microsoft Excel. All patients who participated in the study gave written informed consent after a personal information session at baseline. The study and the use of the patient data were approved by the ethics committee of the University Hospital Erlangen. 

Statistical Analyses:

Statistical analyses were performed using Microsoft Excel (2016, Redmond, WA, USA), SPSS (26, IBM, Armonk, NY, USA) and PRISM-GraphPad (v.9.0.2 Graphpad Holdings, San Diego, CA, USA). Data from the Excel spreadsheet were used to compare and separate data by survey period, sex, age group, and COVID-19 months. It was also used to calculate the respective means and standard deviations. SPSS was used to calculate the statistics. Differences were tested with t-tests and Levene tests. A difference of 10 percentage points (pp) and *p* values < 0.05 were considered statistically significant. The two groups in the cohort were clearly asymmetric, with 350 patients prior to COVID-19 and 39 patients during the COVID-19 pandemic. All available patients were surveyed during the COVID-19 pandemic, so the group was limited by time frame. The power analysis revealed that with a total number of patients of at least 390, with 350 patients in one group and 40 patients in the other, a power of 50% and thus a mean effect size at a two-sided significance level of 5% can be achieved for the distinction between patients without COVID and patients during the COVID-19 pandemic. The specific graphs for each of the categories that were being analyzed were plotted with the help of PRISM-GraphPad.

## 3. Results

This study was performed from May 2010 to June 2021 with 389 patients who agreed to participate, focusing on March 2020 to June 2021, the months in which COVID-19 occurred and was analyzed. In Germany, the first COVID-19 cases were reported in March 2020, and the first wave lasted until May 2020. The second wave lasted from October 2020 to February 2021, the 15-month period when COVID-19 levels were highest in the city of Erlangen, where the study was conducted (Figure 1A). Questionnaires were analyzed in the period from 2010 to 2020 prior to COVID-19 and during the first 15 months (March 2020 to June 2021) of COVID-19. Patients received the questionnaires at different time points during their therapy. The first questionnaire was at day-1 (baseline) before patients started therapy, day 14 (week 2) during the RCT, day 35 (week 5) at the end of the RCT, and day 70 (week 10) immediately before mesorectal surgery. Figure 1B shows the timing of the questionnaires according to the stage of therapy.

The patient cohort consisted of 389 patients with rectal cancer, including 350 patients before COVID-19 and 39 during COVID-19. The proportion of male and female patients before the pandemic was 70.6% and 29.4%, respectively. During the pandemic, the proportion of male patients was 59% and the proportion of female patients was 41%. Most patients were in advanced tumor stages. The proportion of patients at cT3 stage for non-COVID-19 was 65.4% and 65.0% for COVID-19, and the proportion of patients at cT4 stage for non-COVID-19 was 25.5% and 24.0% for COVID-19 (Table 1). The average age was 62.5 years, with the youngest age being 15 years and the oldest 86 years.

The higher the functional score, the better, because it means that the patient has more functional abilities in that category. Functions before the COVID-19 pandemic were compared with those during the COVID-19 pandemic (Figure 2A). Non-COVID-19 patients had the highest mean scores for cognitive functioning (84.1%), physical functioning (78.6%), body image (72.9%), and sexual functioning (72.6%) and the lowest mean scores for future outlook (35.9%) and global health (59.1%) at baseline on day-1 radiochemotherapy (Figure 2A). During the COVID-19 pandemic, functional scores were highest for sexual functioning (84.3%) and cognitive functioning (72.6%) and lowest for future perspective (33.3%) and global health (46.4%). During the COVID-19 pandemic, scores for role function (−17.4 pp), global health status (−12.7 pp), cognitive function (−11.4 pp), physical function (−10.4 pp), and social function (−10.0 pp) decreased the most (*p* < 0.045). Emotional functioning decreased by 7.7 pp; future outlook (−2.6 pp) and body image (0.5 pp) did not change. The only functional score that did not decrease was sexual function, which increased from 72.6% to 84.3%. 

For symptom scores, the higher the score, the greater the burden on the patient. The different symptom scores were compared between the scores prior to the COVID-19 pandemic and during the COVID-19 pandemic (Figure 2B). On day 1, patients had the highest mean scores for fatigue (36.4%), insomnia (32.8%), and diarrhea (32%) and the lowest average scores for micturition problems (15.2%), chemotherapy side effects (7.9%), and nausea and vomiting (6.5%). For COVID-19, patients reported higher scores for most symptoms. The highest scores were reported for physical functioning (51.3%), insomnia (41.9%), and diarrhea (41.9%), and the lowest scores were reported for chemotherapy side effects (8.6%) and constipation (13.2%).

During the COVID-19 pandemic, scores for fatigue (14.9 pp), pain (12.0 pp), and nausea and vomiting (10.6 pp) increased clearly (*p* < 0.047). Weight loss (10.8 pp), diarrhea (9.9 pp), insomnia (9.1 pp), and appetite loss (9.0 pp) tended to increase (*p* < 0.133). The only symptoms that did not increase during this time were dyspnea (3.1 pp), gastrointestinal tract symptoms (2.1), micturition problems (1.6 pp), chemotherapy side effects (0.8 pp), financial difficulties (−5.4 pp), and constipation (−2.4 pp) (*p* > 0.267).

All participating patients received the same questionnaires on day 14 (2 weeks) during the RCT (Figure 3), at the end of the RCT on day 35 (5 weeks) (Figure 4), and just prior to surgery on day 70 (Appendix A). The differences in functional and symptom scores gradually decreased at the three time points. At day 14, none of the COVID-19 scores had worsened by more than 10 points, and only physical functioning (−9.3 pp) and global health status (−9.6) decreased distinctly (*p* < 0.039). At the 35-day time point, the greatest deteriorations were in diarrhea (7.1 pp), physical functioning (−3.2 pp), and cognitive functioning (−3.1 pp). On day 70, there was no further deterioration, body image was only −2.7 pp, and all other functional scores remained unchanged or even improved, such as global health status (11.9 pp). Symptom scores worsened by 12.1 pp for diarrhea, 10.1 pp for constipation, and 8.4 pp for insomnia (*p* < 0.245).

There was a clear time dependence of the functions and symptoms. Therefore, the mean scores of the functions and symptoms were plotted over time. The scores related to sexuality were omitted because there were very few responses here (Figure 5A).

We were interested in whether the differences were attributable to a particular subgroup. Therefore, we compared patients younger than or equal to 64 years of age with older patients and men with women both before and during the COVID-19 pandemic. Functional scores were almost identical in the younger and older subgroups. At baseline, the COVID-19 group had lower functional scores (9.2 pp), which differed little at other time points. The younger patients in the COVID-19 group had a higher symptom burden (30.2%) than the older patients (22.1%) at baseline, which decreased only slightly over time. Men’s functional and symptom scores were only slightly worse in the COVID-19 group at baseline. There were no further differences at subsequent time points. Women’s functional and symptom scores in the COVID-19 group were also worse at baseline, and there were almost no differences at subsequent time points. However, compared with men, women’s scores were significantly worse. In the non-COVID-19 group, women’s functional scores were 10.3 pp lower than in the men’s group at all four time points. In the COVID-19 group, however, women’s functional scores worsened by 10.1 pp to 19.2 pp at the final time point. Like symptom scores, differences in the COVID-19 group increased over time, and women had clearly more symptoms than men.

When looking at the time course, there were clear differences between the individual scores. The main difference is that some functional scores, such as body image in Figure 6 (or chemotherapy side effects, micturition, and defecation behaviors in Appendix A), were absolutely not different over the entire time course. The same is true for symptom scores (Figure 7 and Appendix A). In most cases, the baseline score is worse in the COVID-19 group, and at the next time points, this difference disappears (Figure 6 and Figure 7 1st column). Age does not make as much of a difference. In the domains of pain, financial difficulties, and nausea and vomiting, young patients have significantly more symptoms than older patients (pp > 11.5) (Figure 6 and Figure 7 2nd column).

In contrast to age, sex has a much greater influence. Females in the COVID-19 group have significantly worse functional scores than males in the same group compared with both male and female non-COVID-19 patients. This was true for physical function (mean of the four time points: in men, 1.7 pp; in women, 12.4 pp), role function (m. −0.4 pp, f. 11.1 pp), cognitive function (m. 2.1 pp, f. 9.2 pp), social function (m. 0.1 pp, f. 10.3 pp), and global health (m. 3.8 pp, f. 8.7 pp). Fatigue (m. -0.9 pp, f. 16.6 pp), dyspnea (m. −3.9 pp, f. 14.2 pp), loss of appetite (m. −1.1 pp, f. 8.7 pp), and diarrhea (m. 4.6 pp, f. 8.1 pp) were also clearly more prevalent in women (Figure 6 and Figure 7 3rd column).

## 4. Discussion

Patients with rectal cancer suffer from various symptoms, such as diarrhea, constipation, pain (peripheral and abdominal), weight loss, and many others, during all stages of treatment. All patients, regardless of age and sex, reported extremely low functional scores and high symptom scores during radiochemotherapy (day 35), indicating that patients’ lives and well-being changed dramatically, which could be due to radiochemotherapy. Patients responded differently to therapy. During the RCT, patients reported very high levels of fatigue and suffered from various side effects of chemotherapy. The intensity of symptoms varied greatly at each stage, but this may be caused or influenced by other factors. 

In most patients receiving radiochemotherapy, quality of life (QoL) scores worsened during the different stages of therapy. There was little difference between the two age groups studied, with either the older group performing better than the younger or vice versa, demonstrating that age does not have a major impact on patients’ functional abilities and scores [5]. Symptom scores were almost identical in older and younger patients, with a few exceptions. Pain, insomnia, and nausea and vomiting were significantly more prevalent in young pandemic patients. This reflects the greater life experience and resilience of older individuals. Similarly, an older individual is more likely to have been affected by cancer and therefore more likely to have dealt with the possibility of developing the disease. Here, however, only the young pandemic patients had higher symptoms, which may show that COVID-19 stress triggered an overdose. Financial difficulties were particularly important, as younger patients had a higher overall burden. This is understandable, as young patients are still living their lives to the fullest and their future is not yet financially secure, so they would be severely affected by any COVID-19-related financial constraints (personal or government expenses). What is surprising here is that the older pandemic patients reported almost no financial difficulties and that there was probably a pandemic-related realization that there are more important things than finances. In a previous study, we had not seen these effects as strongly [5]. The functional scores for younger and older patients were almost identical. In the young pandemic group, there was a clear outlier in most scores at the last time point. This is a limitation of this study because there were only a limited number of patients during this time period. However, since no additional patients were treated, this problem could not be resolved.

Although age did not have a major effect on patients, sex differences had a notable effect on functional and symptom scores. Female patients reported lower functional scores and higher symptom scores before and during the pandemic than male patients. Females were particularly affected at baseline during the pandemic. Particularly in terms of physical function and social function, women experienced additional distress during the pandemic. In terms of symptoms, women were also particularly burdened by fatigue, loss of appetite, and shortness of breath during the pandemic. This suggests that female patients were less affluent and had higher levels of stress at all phases and times of the surveys. In a study of sex differences in neoadjuvant radiochemotherapy of the rectum, no differences in radiosensitivity per se were found. However, women received slightly higher energy doses of ionizing radiation relative to body mass. Women’s quality of life tended to be impaired, probably because of chemotherapy during radiochemotherapy [6]. We did not assess whether impairment in quality of life was associated with a shortening of life. In a previous work, we showed in the same rectal cancer cohort that there is a significant reduction in survival for individuals reporting increased fatigue, pain, and loss of appetite, among other symptoms [7]. Here, we did not test this association because the follow-up period for the COVID-19 cohort was too short.

A very interesting observation is that the baseline was generally much worse during the pandemic. This means that the pandemic caused an even higher burden, which then decreased during therapy to the level of the patients before the pandemic. This means that during therapy, the burden caused by COVID-19 was displaced from everyday therapy.

At the baseline of radiochemotherapy, patients typically feel very fatigued and tired, making it difficult for them to perform their daily tasks. A total of 60% of patients reported weakness during and after treatment, and 30% reported fatigue for years after therapy [8]. Fatigue was also significantly increased in the pandemic patients in this study at baseline in both young and elderly patients and in both men and women. Thus, women during the pandemic were additionally more fatigued than the already more fatigued women before the pandemic over the entire period. Problems with micturition are another symptom that increases in patients undergoing RCT, as do other symptoms, such as inability to urinate, urinary urgency, fever, and chills. These medications cause irritation of the bladder lining, which can lead to inflammation and bleeding. The drugs can also cause nerve damage, resulting in pain or burning during urination and frequent urination [9]. However, there was no increase in micturition problems in the pandemic cohort.

Functional abilities may also decrease due to the therapeutic effects of radiochemotherapy. It is conceivable that these symptoms increased and worsened during the pandemic because there were fewer distractions for patients, more loneliness and stress, and increased fatigue and emotional stress, which in turn affected patients’ overall mood. Women in the pre-pandemic cohort reported poorer body image. Surprisingly, however, women in the pandemic group did not. This perhaps suggests that attention has shifted to other important things, making body image less important. Often, patients tend to be dissatisfied with their body image after receiving a stoma [10]. However, we did not record this time point anymore.

When comparing the clinical characteristics of different patients, one of the most important aspects or characteristics would be the stage of the disease at which the patients were diagnosed. Originally, it was thought that patients were highly likely to be affected by their stage of cancer and that this could lead to greater functional difficulties and symptoms. From the results, it can be inferred that although the number of patients in the groups differed before and during COVID-19 treatment, the percentage of patients diagnosed with a specific stage of disease was quite similar in both groups. This suggests that disease stage had little effect on the functional difficulties and symptoms and overall well-being of the patients in this study. 

There were several limitations, mainly reducing the number of social contacts during the pandemic. Patients were stressed, anxious, and irritable, and their emotional performance decreased sharply. Because there were so many restrictions on social contact, patients had a very limited and controlled number of people visiting or accompanying them [11]. Cancer patients may have a weakened and sensitive immune system, making them more susceptible to infection with COVID-19. In addition, elderly patients are even more at risk, especially if they are also treated with chemotherapy [12]. Most patients were worried for these reasons, and there was also a lot of uncertainty about the post-treatment sessions, which led to more anxiety and depression among patients. For working parents with young children, the shutdowns were problematic, as they had to balance their work, health, and personal lives [13]. 

The pandemic was also a stressful time for hospital staff because of the lack of protective clothing. The working hours and workload were high, there was no childcare support, job satisfaction decreased, and many other difficulties occurred [14]. All of these problems may have also indirectly affected the patients, as they also experience stress in the hospitals on a regular basis, which in turn increases their stress level and uncertainty about their recovery and the next steps in their therapy. Of course, there are several limitations in such a study of an unanticipated pandemic. Several factors may affect quality of life that we did not anticipate and therefore did not study. One issue is whether there is a change in patient support through counseling or care. Retrospective inquiries revealed no evidence of a significant change. Another limitation is that we did not collect the menstrual status of the patients, which could have a significant impact on the QOL.

When comparing functional and symptom scores before and during the pandemic, the differences between the sexes were very large. Female patients are generally more attentive when it comes to health and illness. The female sex is more understanding and attentive to their illnesses, “Female patients are vocal and active participants in doctor–patient communication” [15]. Male patients tend to leave decisions to physicians and participate less in the decision-making process. Therefore, it is easier to understand why female patients reported a greater decrease in functional scores and an increase in symptom scores, as they are more attentive and cautious when it comes to health issues. The pandemic would have only exacerbated this as emotional distress, symptoms, and side effects set in. Female patients may have been more aware of these issues than male patients, causing the notable difference in scores before and during the pandemic in both sexes.

## 5. Conclusions

The patients’ quality of life was clearly affected and worsened by the COVID-19 pandemic. There were several differences that revealed that most functions and symptoms were less favorable during the pandemic. Although it was hypothesized that age would have an effect on function and symptom scores, no large difference was found between age groups. There was a small difference between pre-pandemic and pandemic, with slightly worse scores during the pandemic. Sex was a very important aspect, as women had lower functional scores and higher symptoms than men both before the pandemic and during the pandemic. Cancer patients suffer from numerous problems and limitations at all stages of therapy, but especially during radiochemotherapy, when the body suffers the most, which was exacerbated during the pandemic.

## Figures and Tables

**Figure 1 healthcare-10-01513-f001:**
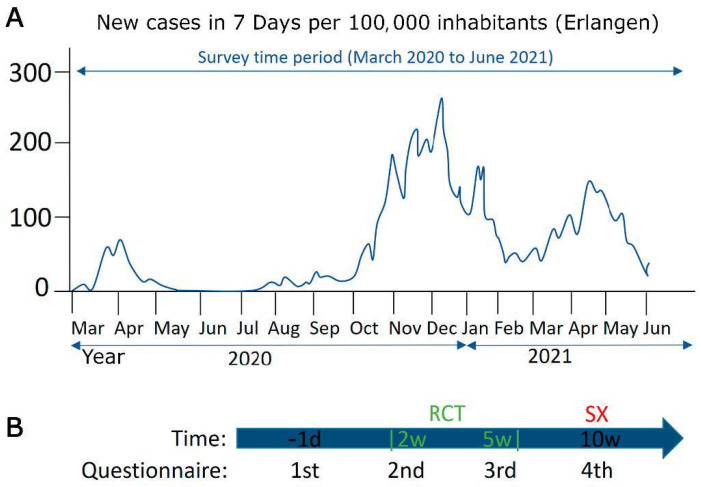
(**A**) The number of new cases in seven days per 100,000 inhabitants in Erlangen from March 2020 to June 2021. (**B**) The timeline corresponds to the period in which the patient received the questionnaire and in which therapy stage the patient was at (RCT = radiochemotherapy; SX = surgery).

**Figure 2 healthcare-10-01513-f002:**
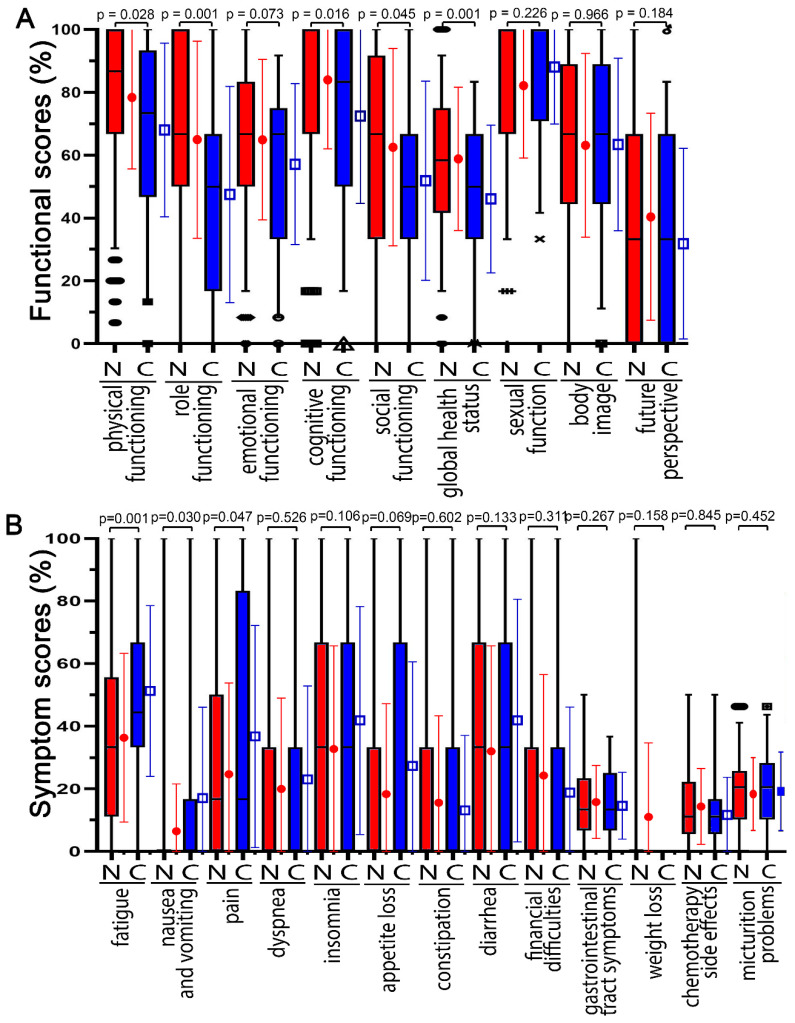
The functional and symptom score distribution of prior to the COVID-19 pandemic (N = non COVID) compared with during the COVID-19 pandemic (C = during COVID pandemic). Baseline values were surveyed on day 1 before radiochemotherapy. (**A**) Functional scores are: physical functioning, role functioning, emotional functioning, cognitive functioning, social functioning, global health status, sexual function, body image, and future perspective. (**B**) Symptom scores are: fatigue, nausea and vomiting, pain, dyspnea, insomnia, appetite loss, constipation, diarrhea, financial difficulties, gastrointestinal tract symptoms, weight loss (WL), chemotherapy side effects, and micturition problems. The different symbols signify the outliers and extreme values. □ and ● indicate the mean and error bars the standard deviation.

**Figure 3 healthcare-10-01513-f003:**
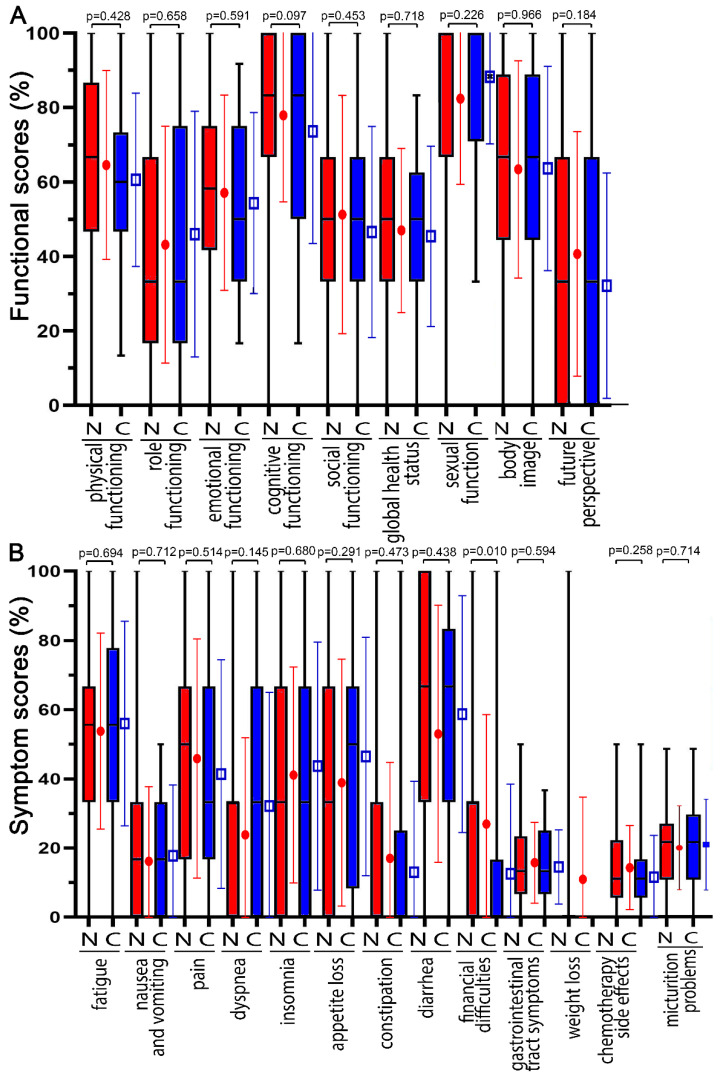
The functional and symptom score distribution of prior to COVID-19 pandemic (N = non COVID) compared with during COVID-19 pandemic (C = during COVID pandemic). Scores were surveyed on day 14 (2 weeks) during radiochemotherapy. (**A**) Functional scores are: physical functioning, role functioning, emotional functioning, cognitive functioning, social functioning, global health status, sexual function, body image, and future perspective. (**B**) Symptom scores are: fatigue, nausea and vomiting, pain, dyspnea, insomnia, appetite loss, constipation, diarrhea, financial difficulties, gastrointestinal tract symptoms, weight loss, chemotherapy side effects, and micturition problems. □ and ● indicate the mean and error bars the standard deviation.

**Figure 4 healthcare-10-01513-f004:**
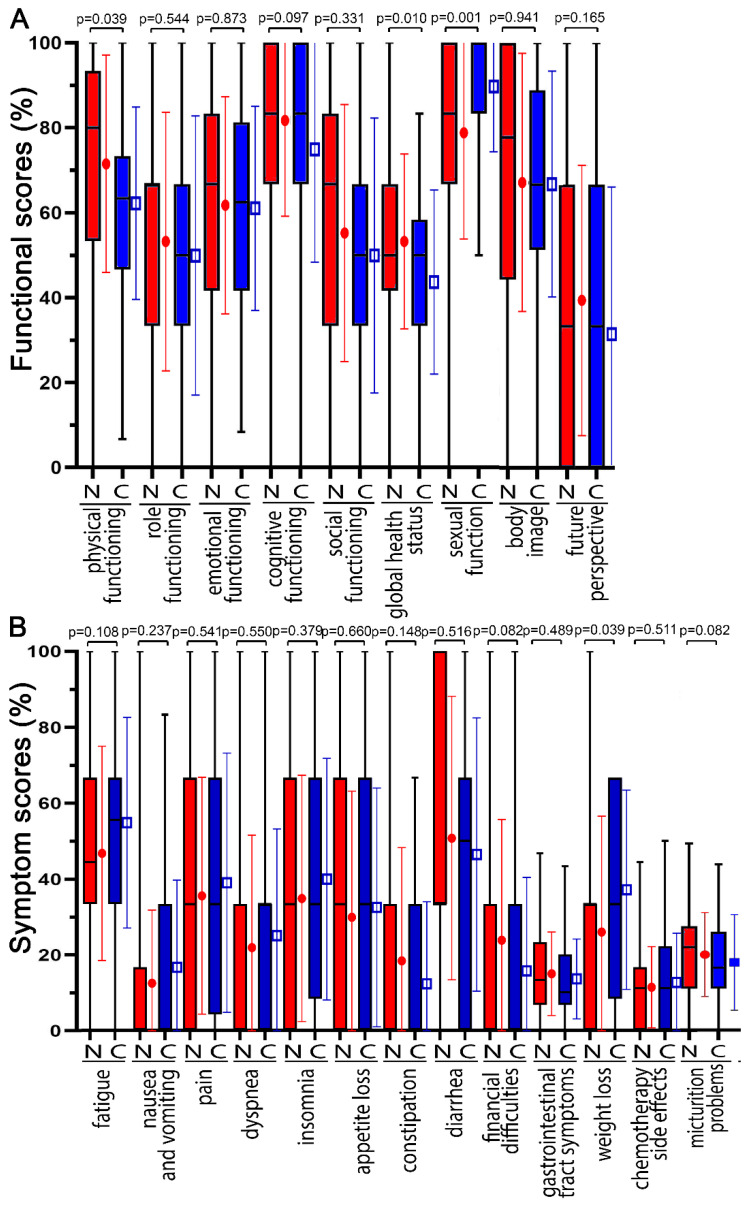
The functional and symptom score distribution of prior to COVID-19 pandemic (N = non COVID) compared with during COVID-19 pandemic (C = during COVID pandemic). Scores were surveyed on day 35 (5 weeks) at the end of radiochemotherapy. (**A**) Functional scores are: physical functioning, role functioning, emotional functioning, cognitive functioning, social functioning, global health status, sexual function, body image, and future perspective. (**B**) Symptom scores are: fatigue, nausea and vomiting, pain, dyspnea, insomnia, appetite loss, constipation, diarrhea, financial difficulties, gastrointestinal tract symptoms, weight loss, chemotherapy side effects, and micturition problems. □ and ● indicate the mean and error bars the standard deviation.

**Figure 5 healthcare-10-01513-f005:**
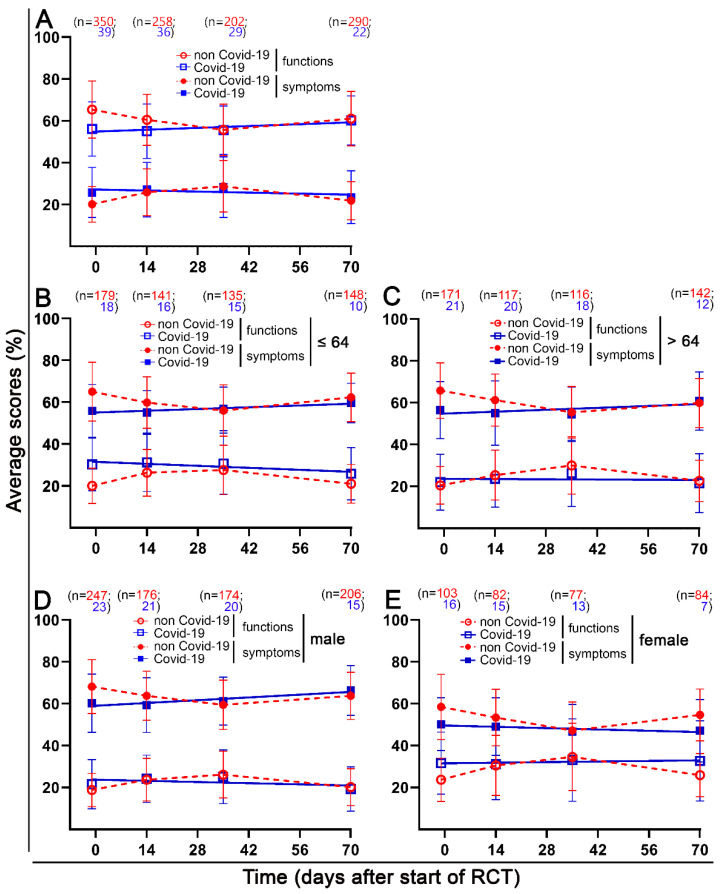
Time course of different functional and symptom scores, the functional and symptom scores for the two age groups (age <64 and age >64), and two sexes (male and female) at day 1, day 14, day 35, and day 70 at pre-pandemic times and pandemic times. (**A**) The general functions and symptoms for both the pre-pandemic and pandemic periods. (**B**) The young age cohort ≤ 64 years compared to the older age cohort (**C**) >64 years before the pandemic and in the pandemic period. (**D**) Males compared to (**E**) females before the pandemic and in the pandemic period. Error bars indicate the standard deviation.

**Figure 6 healthcare-10-01513-f006:**
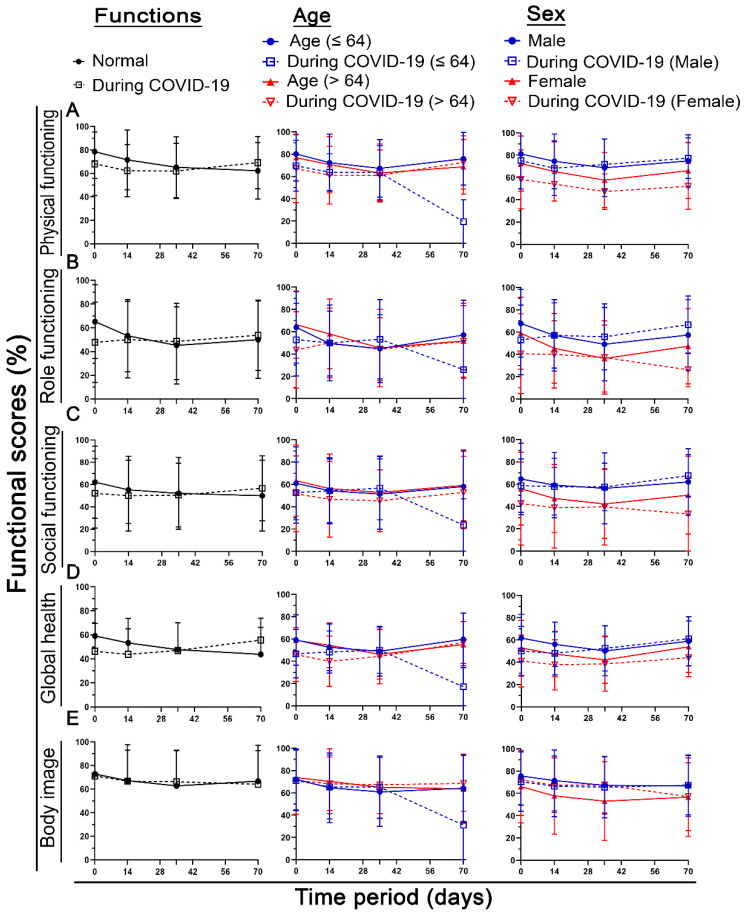
Time course of different functional scores at day 1, day 14, day 35, and day 70. In the first column, the entire cohort is separated into normal (before COVID-19) and during COVID-19. In the second column, the cohort was divided into patients younger or 64 to older than 64. In the third column, females and males of the cohort were compared for (**A**) physical function, (**B**) role function, (**C**) social function, (**D**) global health status, and (**E**) body image. Error bars indicate the standard deviation.

**Figure 7 healthcare-10-01513-f007:**
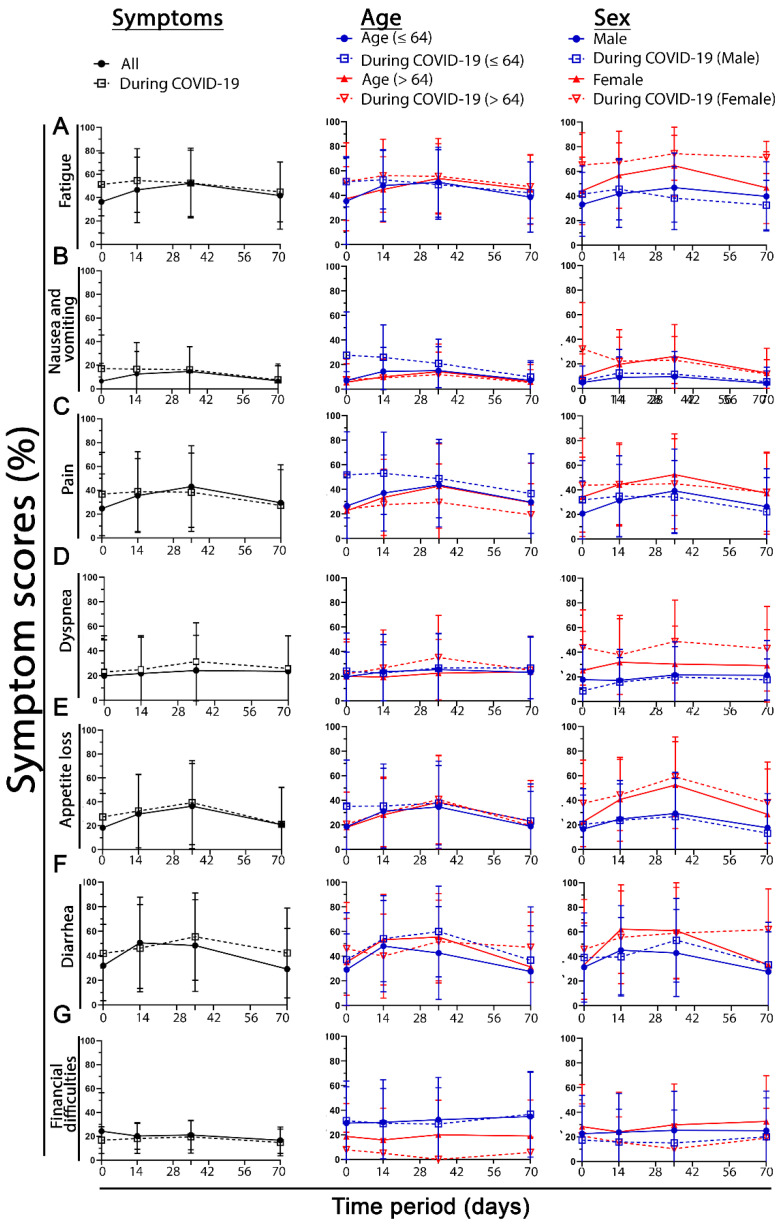
Time course of different symptom scores at day 1, day 14, day 35 and day 70. In the first column the entire cohort is separated into normal (before COVID-19) and during COVID-19. In the second column the cohort was divided into patients younger or 64 to older than 64. In the third column females and males of the cohort were compared for (**A**) fatigue, (**B**) nausea and vomiting, (**C**) pain, (**D**) dyspnea, (**E**) appetite loss, (**F**) diarrhea, and (**G**) financial difficulties at the different time periods at which the surveys took place, at day 1, day 14, day 35, and day 70. Error bars indicate the standard deviation.

**Table 1 healthcare-10-01513-t001:** Patient characteristics of the non-COVID-19 and the COVID-19 cohort.

Variable		Non COVID-19 (%)	COVID-19 (%)
Rectal cancer patients		350 (90.0)	39 (10.0)
Sex (male/female)		247/103 (70.6/29.4)	23/16 (59.0/41.0)
Age ≤64/>64		179/171 (51.1/48.9)	18/21 (46.2/53.8)
Tumor grade 1; 2; 3;		13 (3.7) 276 (79.0) 61 (17.3)	0 (0) 30 (76.7) 9 (23.3)
Patients without surgery		29 (8.3)	3 (8.0)
Stage	cT 2; 3; 4;	32 (9.2); 229 (65.4); 89 (25.5)	5 (12.0); 25 (64.0); 9 (24.0);
	pT 0; 1; 2; 3; 4;	40 (12.5); 30 (9.3); 90 (28); 134 (41.8); 27 (8.4)	8 (20.8); 2 (4.2); 11 (29.2); 15 (41.7); 2 (4.2);
	cN 0; 1; 2;	81 (23.3); 188 (53.8); 80 (23.0);	10 (25.0); 16 (41.7); 13 (33.3);
	pN 0; 1; 2;	242 (75.3); 72 (22.5); 36 (11.3);	29 (79.2); 6 (16.7); 2 (4.2);
	cM 0; 1;	279 (79.6); 71 (20.4);	30 (77.8); 9 (22.2);
	cUICC 1; 2; 3; 4;	16 (4.7) 55 (15.7) 205 (58.5) 74 (21.1)	4 (10.7) 3 (7.1) 24 (60.7) 8 (21.4)
	pUICC 1; 2; 3; 4;	102 (29.1) 107 (30.6) 87 (24.9) 54 (15.5)	16 (42.1) 10 (26.3) 6 (15.8) 6 (15.8)

## Data Availability

The datasets used and analyzed during the current study are available from the corresponding author on reasonable request.

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
