# Peer review of "Substantial Impairment of Quality of Life during COVID-19 Pandemic in Patients with Advanced Rectal Cancer"

_healthcare, 2022, doi:10.3390/healthcare10081513_

Round 1
Reviewer 1 Report
The work is weakly written overall, with the exception of the introduction and the abstract. Way many errors and repetitions may be found everywhere. Although the intention and work put into presenting the outcome appear to be commendable, the presentation's quality is low that I will not be able to evaluate it unless it is significantly changed. Even after paying close attention to the figures, it is difficult to see how this differs from other examples. Before resubmitting this article for another round of reviews, I advise the authors to either get one of their native-speaking colleagues to proofread it or employ a proofreading service. Also, increase the quality of the figures while keeping the readers in mind.
Author Response
We have had the entire manuscript linguistically revised. We revised the illustrations and, among other things, chose uniform colors for the different groups. We hope that this will make the work easier to understand.
Reviewer 2 Report
Manuscript Title: Clearly impaired quality of life in the treatment of advanced rectal cancer during the Covid-19 pandemic
Manuscript Summary: In this article, Dennison, et al., analyzed the impact of COVID 19 pandemic on quality of life of patients with advanced rectal cancer. The authors conducted qualitative survey and concluded that sex (females more than males) than age played a major role in perception of functional abilities and symptoms, which was further impacted by COVID 19. The manuscript is straightforward in presenting the findings. However, there are several areas where it requires the authors to provide additional information to clarify the findings presented in the article. The concerns are listed below:
Major Concerns:
1. The title of the paper is confusing and needs minor rewording to accurately match with the results presented in the article.
2. In the introduction section, the authors have not conveyed adequately the significance of the study nor its impact to better the quality of life of patients with advanced rectal cancer.
3. The article has not made it clear if the decrease in functional scores and increase in symptoms had a detrimental effect on patient survival (specifically for females) pre and during COVID-19 pandemic.
4. The number of patients included part of the study following COVID-19 pandemic is only 10% of the total participants included in the study. The authors have to clarify the statistical parameters (power analysis if possible) used for the analysis in the study. Please specify this in the materials and methods.
5. The authors have to make it clear if the menstrual status or tumor grade of the females used in the study have an impact on functional scores.
6. The authors have not indicated if there is a difference in counselling therapy availability for patients’ pre and during COVID 19 and if it has any impact on increasing the functional score of the patients.
Minor Concerns:
1. Figure legend 2 is the only one italicized in the main manuscript. Please keep the formatting consistent throughout the article.
2. Figure 5, panel C and E, the differences between the curves between age groups and males and females are not clear especially the color light green.
3. In Figure 5, why are the data for patient cohort ≤ 64 (panel C) and for males (panel E) represented again when they are already represented in panels B and D. The data presented is really confusing and difficult to interpret.
Round 2
Reviewer 1 Report
The authors have addressed the concern I raised regarding the quality of the presentation as well as the quality of the figures. I still see some grammatical errors, for example, see the first paragraph in the introduction where the word "were" is typed twice. However, errors like these can be improved during the proofreading stage. Overall, I recommend the publication of the revised article.
Reviewer 2 Report
The authors have clarified and made necessary changes to the manuscript based on the concerns raised. As a result, the manuscript can be accepted in its present form.